# Long-Term Survival of Three Fungal Species in Lyophilized Human Blood Plasma

Omar Anwar Elkadi,[a,b] Mervat Elanany,[c] Mohammed A. Ramadan[a]

aDepartment of Microbiology and Immunology, Faculty of Pharmacy, Cairo University, Cairo, Egypt
bDar Elsalam Cancer Center, Ministry of Health and Population, Cairo, Egypt
cDepartment of Clinical Pathology, Faculty of Medicine, Cairo University, Cairo, Egypt

Mervat Elanany and Mohammed A. Ramadan should be considered joint senior authors.

**KEYWORDS** *Aspergillus flavus*, *Aspergillus niger*, *Penicillium chrysogenum*, diagnostics, lyophilization, medical mycology

The low prevalence of some diseases, such as invasive fungal diseases, limits the availability of clinical specimens needed for developing and testing novel diagnostics for such diseases. Thus, the development and testing of such diagnostics usually require the use of banked specimens (1). However, the use of such specimens mandates the evaluation of the impact of the storage conditions on the stored specimens (1).

Lyophilization allows the long-term storage of biological samples at room temperature and lowers the cost of storage and shipping, which makes it a valuable method for biobanking (2, 3). Moreover, lyophilization improves the performance of some spectroscopic methods such as infrared spectroscopy (4). In a previous study, plasma samples spiked with *Aspergillus* and *Penicillium* species were lyophilized and then analyzed by infrared spectroscopy to study the potential of infrared spectroscopy as a blood-based diagnostic tool for invasive aspergillosis (4). As the survival of microorganisms in samples stored in a biobank can affect the assay being investigated as a novel diagnostic, this study reports the survival of these fungal species in these lyophilized plasma samples after 4 years of storage at room temperature.

Samples used in this study were prepared in a previous study, as follows (4). The mycelia of clinical isolates of *Aspergillus flavus*, *Aspergillus niger*, and *Penicillium chrysogenum* were used to aseptically spike fresh-frozen plasma in blood tubes (1 ml), and the samples were then lyophilized overnight at −50°C to be used in the study. These isolates were identified to the species level by Sanger sequencing and matrix-assisted laser desorption ionization–time of flight (MALDI-TOF) identification in addition to their macroscopic and microscopic morphology; the isolates were subcultured on Sabouraud dextrose agar (SDA) at 30°C for 7 days, which resulted in adequate mycelial development to be used for spiking the samples (4). The remaining lyophilized plasma samples in addition to a nonspiked lyophilized plasma sample, as a negative control, were stored at room temperature in the dark until the time of conducting this study.

After 4 years, Sabouraud dextrose broths (SDBs) were aseptically spiked by a few milligrams of the stored lyophilized samples (directly, without prior reconstitution) and then incubated at 30°C for 2 to 3 days. Once the mycelia were visible, part of the mycelia was transferred onto SDA and then incubated at 30°C and examined daily for typical colonies for 7 days. Colonies from the plate and mycelia from growth in SDB were also examined under a microscope for characteristic features of the fungi.

Microscopic examination included examining the colonies on the plate itself with a low-power lens. To examine the fungi by high-power and oil immersion lenses, an agar plug from the SDA with the corner of the colony and/or mycelial growth from SDB was transferred to a flame-sterilized slide.

Address correspondence to Omar Anwar Elkadi, Omar.elkadi@live.com.

Three fungi have been recovered from lyophilized plasma after four years of storage at room temperature

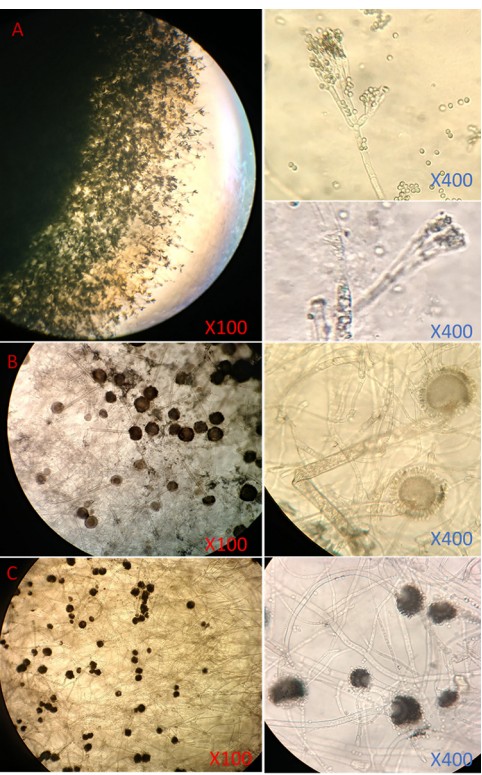

**FIG 1** Microscopic morphology of *Penicillium chrysogenum* (A), *Aspergillus niger* (B), and *Aspergillus flavus* (C).

Within 2 days, the three fungal species had grown into visible hyphal masses in SDB, indicating the survival of the fungi, while the negative control showed no growth, which reflects the efficacy of the aseptic conditions employed in this study. After the transfer of these hyphal masses onto SDA, the three fungi were identified by their typical colonies within 5 days, confirming that the surviving fungi are the fungi originally spiked in the plasma prior to lyophilization. All the fungal species showed a flat topography. The colony of *P. chrysogenum* was velvety, while the colonies of *A. flavus* and *A. niger* were granular. Both *A. niger* and *P. chrysogenum* colonies were initially white and then became black and blue-green, respectively, while *A. flavus* colonies were initially yellow and then became yellowish green.

Microscopically, *A. flavus* was identified by its rough conidiophores, radiating conidial heads, and phialides that cover the entire surface of its globose vesicles. *A. niger* was identified by its smooth conidiophores and radiating black conidial heads with metulae twice the length of the phialides, while the *Penicillium* species showed their characteristic brush-like conidiophores and no vesicles (Fig. 1).

This report reflects the three fungal species' survival in blood plasma, survival during lyophilization, and survival during long-term storage, which are consistent with previous studies that demonstrated each of these features using the same or other microorganisms. For instance, the ability of *Aspergillus* species to grow in blood serum has been reported in a previous study (5). The survival of bacteria in plasma after lyophilization has also been previously reported (6). The long-term survival of microorganisms after freeze-drying is well documented, which makes it a commonly used method for the preservation of microorganisms (7). However, as the samples were prepared in a previous study and the initial count was not recorded, it was not possible to estimate the exact survival rate.

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
