## [Reviewer comments · Microbiology Spectrum]

Microbiology Spectrum

Long Term Survival of Three Fungal Species in Lyophilized Human Blood Plasma

Omar Elkadi, Mervat Elanany, and Mohammed Ramadan

Corresponding Author(s): Omar Elkadi, Dar elsalam Cancer center

Review Timeline:

Submission Date:	May 6, 2021
Editorial Decision:	June 5, 2021
Revision Received:	June 5, 2021
Accepted:	June 13, 2021

Editor: N. Esther Babady

Reviewer(s): The reviewers have opted to remain anonymous.

Transaction Report:

DOI: <https://doi.org/10.1128/Spectrum.00222-21>

June 5, 2021

Mr. Omar Anwar Elkadi
Dar elsalam Cancer center
El-Malek El-Saleh, Al Kafour, Old Cairo,
Cairo
Egypt

Re: Spectrum00222-21 (Long Term Survival of Three Fungal Species in Lyophilized Human Blood Plasma)

Dear Mr. Omar Anwar Elkadi:

Thank you for submitting your manuscript to Microbiology Spectrum. While your study is of interest, given the limited number of fungal strains evaluated, it does not warrant publication as a full research article. Please consider using the new data letter format and revise the paper along the lines suggested by the reviewers. When submitting the revised version of your paper, please provide (1) point-by-point responses to the issues raised by the reviewers as file type "Response to Reviewers," not in your cover letter, and (2) a PDF file that indicates the changes from the original submission (by highlighting or underlining the changes) as file type "Marked Up Manuscript - For Review Only". Please use this link to submit your revised manuscript - we strongly recommend that you submit your paper within the next 60 days or reach out to me. Detailed information on submitting your revised paper are below.

Link Not Available

Sincerely,

N. Esther Babady

Journals Department
Reviewer comments:

Reviewer #1 (Comments for the Author):

Elkadi et al present a nice report about reviving fungi from lyophilized state in a plasma matrix. The findings have value for a number of applications. I have only some minor comments to further refine this well-written report:

Did the authors prepare the fungi in a special way prior to lyophilization? For instance, did they assure sporulation was happening, or a certain level of sporulation? Even if this was described in cited work, please share again here briefly for completeness.

Line 35: were the samples reconstituted first, or was the lyophilized powder used as the inoculum itself?

Reviewer #2 (Comments for the Author):

While this an important step in assessing if lyophilization is a useful way to preserve this type of specimen this study is very limited in its evaluation (only one strain each of three fungal species with no replicates) and does not measure performance against any other method of preservation.

The authors demonstrated the limited number of fungi were still viable after 4 years of storage however it is unclear how this compared (or if there is a benefit) to other storage methods. The authors state this method offers potentially "improved stability" (line 15) but no comparator evaluation or data are provided. If the goal of the study was to identify a way to preserve actual clinical specimens (in this case plasma, as well as the pathogen) for later use in diagnostic assay development/performance evaluation (line 12) there should also be some method of evaluation to determine if reconstituted plasma has similar properties as fresh specimen.

This study is very limited in scope in its current form.

Staff Comments:

Preparing Revision Guidelines

- Point-by-point responses to the issues raised by the reviewers in a file named "Response to Reviewers," NOT IN YOUR COVER LETTER.
- Upload a compare copy of the manuscript (without figures) as a "Marked-Up Manuscript" file.
- Each figure must be uploaded as a separate file, and any multipanel figures must be assembled

into one file.

- Manuscript: A .DOC version of the revised manuscript
- Figures: Editable, high-resolution, individual figure files are required at revision, TIFF or EPS files are preferred

For complete guidelines on revision requirements, please see the Instructions to Authors at [link to page]. **Submissions of a paper that does not conform to Microbiology Spectrum guidelines will delay acceptance of your manuscript.**

Please return the manuscript within 60 days; if you cannot complete the modification within this time period, please contact me. If you do not wish to modify the manuscript and prefer to submit it to another journal, please notify me of your decision immediately so that the manuscript may be formally withdrawn from consideration by Microbiology Spectrum.

If you would like to submit an image for consideration as the Featured Image for an issue, please contact Spectrum staff.

Spectrum00222-21

Review Elkadi et al.

While this an important step in assessing if lyophilization is a useful way to preserve this type of specimen this study is very limited in its evaluation (only one strain each of three fungal species with no replicates) and does not measure performance against any other method of preservation.

The authors demonstrated the limited number of fungi were still viable after 4 years of storage however it is unclear how this compared (or if there is a benefit) to other storage methods. The authors state this method offers potentially "improved stability" (line 15) but no comparator evaluation or data are provided. If the goal of the study was to identify a way to preserve actual clinical specimens (in this case plasma, as well as the pathogen) for later use in diagnostic assay development/performance evaluation (line 12) there should also be some method of evaluation to determine if reconstituted plasma has similar properties as fresh specimen.

This study is very limited in scope in its current form.

Dear Dr. Babady,

We are grateful to the reviewers for their insightful comments on our paper. We have been able to incorporate changes to address most of the reviewers' comments, which are highlighted in the "Marked Up Manuscript" pdf in blue for those corresponding to the comments of reviewer #1 and red for those of reviewer#2. A point-by-point response to the reviewers' comments and concerns is listed below.

Comments from Reviewer#1:

1. *Did the authors prepare the fungi in a special way prior to lyophilization? For instance, did they assure sporulation was happening, or a certain level of sporulation? Even if this was described in cited work, please share again here briefly for completeness.*

The details of subculturing the fungi have been briefly described at lines 20-22.

2. *Line 35: were the samples reconstituted first, or was the lyophilized powder used as the inoculum itself?*

We have further elaborated that it was used "directly, without prior reconstitution" at line 26.

Comments from Reviewer#2:

1. *The authors demonstrated the limited number of fungi were still viable after 4 years of storage however it is unclear how this compared (or if there is a benefit) to other storage methods.*

This study was not intended to be a comparative study nor creating a new storage method, we were just reporting the survival of the fungi in lyophilized human blood plasma in samples prepared in one of our previous studies four years ago, and we were just elaborating the cases where such samples can be encountered/created. For that purpose, we only submitted the manuscript for publication as an observation (and has been modified as a new data letter as requested by the editor) as we have noticed that this has not been reported before. We have modified the statement about the purpose of the study in line 13 to emphasize that we are only reporting this observation.

2. *The authors state this method offers potentially "improved stability" (line 15) but no comparator evaluation or data are provided.*

In this sentence (now in line 6 after the amendments), we were just mentioning the advantages of lyophilization that was reported in previous studies (see references 2 and 3) to elaborate why plasma samples can be encountered in lyophilized forms. We have rephrased the statement to make it clearer.

3. *If the goal of the study was to identify a way to preserve actual clinical specimens (in this case plasma, as well as the pathogen) for later use in diagnostic assay development/performance evaluation (line 12) there should also be some method of evaluation to determine if reconstituted plasma has similar properties as fresh specimen.*

We agree with the reviewer that other aspects should also be considered in diagnostic assay development/performance evaluation, but as mentioned in the first comment, we were just reporting the survival of the fungi in lyophilized human blood plasma in samples prepared in one of our previous studies four years ago, which is one of the aspects that also can have an impact on diagnostic assay development/evaluation. "identifying a way to preserve clinical specimens" is not in the goal of the study, we were just mentioning when such samples can be encountered.

Other Changes:

To modify the manuscript into the new data letter format, the abstract, the headings, and the conclusion have been removed, some sentences have been trimmed, and figure 1 has been removed.

June 13, 2021

Mr. Omar Anwar Elkadi
Dar elsalam Cancer center
El-Malek El-Saleh, Al Kafour, Old Cairo,
Cairo
Egypt

Re: Spectrum00222-21R1 (Long Term Survival of Three Fungal Species in Lyophilized Human Blood Plasma)

Dear Mr. Omar Anwar Elkadi:

Your manuscript has been accepted, and I am forwarding it to the ASM Journals Department for publication. You will be notified when your proofs are ready to be viewed.

Sincerely,

N. Esther Babady
Editor, Microbiology Spectrum
